# Cross−Talk between Transcriptome Analysis and Physiological Characterization Identifies the Genes in Response to the Low Phosphorus Stress in *Malus mandshurica*

**DOI:** 10.3390/ijms23094896

**Published:** 2022-04-28

**Authors:** Hong Zhao, Yawei Wu, Luonan Shen, Qiandong Hou, Rongju Wu, Zhengchun Li, Lin Deng, Xiaopeng Wen

**Affiliations:** 1Key Laboratory of Plant Resource Conservation and Germplasm Innovation in Mountainous Region (Ministry of Education), Institute of Agro-Bioengineering/College of Life Sciences, Guizhou University, Guiyang 550025, China; zhaohonggzu@163.com (H.Z.); qiandhou@163.com (Q.H.); rongju_wu@163.com (R.W.); 2Institute of Pomology Science, Guizhou Academy of Agricultural Sciences, Guiyang 550006, China; yaweiwu2006@163.com; 3College of Forestry, Guizhou University/Institute for Forest Resources & Environment of Guizhou, Guiyang 550025, China; sln1917470278@163.com (L.S.); zcliah@163.com (Z.L.); d846129692@163.com (L.D.)

**Keywords:** *Malus mandshurica*, physiological character, transcriptome analysis, low−Pi stress, Pi transporter

## Abstract

Phosphorus (Pi) is a macronutrient essential for plant growth, development, and reproduction. However, there is not an efficient available amount of Pi that can be absorbed by plants in the soil. Previously, an elite line, MSDZ 109, selected from *Malus mandshurica*, was justified for its excellent tolerance to low phosphorus (low−Pi) stress. To date, however, the genes involved in low−Pi stress tolerance have not yet been unraveled in this species. Currently, the physiological responses of this line for different days to low−Pi stress were characterized, and their roots as well as leaves were used to carry out transcriptome analysis, so as to illuminate the potential molecular pathways and identify the genes involved in low−Pi stress−response. After exposure to low−Pi treatment (32 µmol/L KH_2_PO_4_) for 20 day after treatment (DAF) the biomass of shoots was significantly reduced in comparison with that of the stress−free (control), and root architecture diversely changed. For example, the root growth parameters e.g., length, surface area, and total volume somewhat increase in comparison with those of the control. The activity of acid phosphatase (ACP) increased with the low−Pi treatment, whereas the photosynthetic rate and biomass were declining. The activity of antioxidant enzymes, e.g., superoxide dismutase (SOD), peroxidase (POD), and catalase (CAT), were substantially elevated in response to low−Pi treatment. Many enzyme−related candidate genes e.g., *MmCAT1*, *MmSOD1* and *MmPOD21* were up−regulated to low−Pi treatment. Furthermore, Kyoto Encyclopedia of Genes and Genomes (KEGG) pathway analysis indicated that the processes of photosynthesis, plant hormone signal transduction, and MAPK signaling pathway were affected in the low−Pi response. In combination with the physiological characterization, several low−Pi−responsive genes, e.g., PHT, PHO, were identified, and the genes implicated in Pi uptake and transport, such as *MmPHT1;5*, *MmPHO1*, *MmPAP1*, etc., were also obtained since their expression status varied among the exposure times, which probably notifies the candidates involved in low−Pi−responsive tolerance in this line. Interestingly, low−Pi treatment activated the expression of transcription factors including the WRKY family, MYB family, etc. The available evidences will facilitate a better understanding of the roles of this line underlying the high tolerance to low−Pi stress. Additionally, the accessible data are helpful for the use of the apple rootstock *M. mandshurica* under low−Pi stress.

## 1. Introduction

Phosphorus (Pi) is an essential macronutrient that is a requisite for the growth, development, and reproduction of plants [1]. Although the total amount of Pi in soils is abundant, there is not much available Pi that can be absorbed by plants due to high fixation and slow diffusion [2,3,4]. Therefore, improvement of the absorption and utilization of Pi is particularly important to attenuate the Pi deficiency of crops [5]. To cope with the shortage of available Pi in soil, plants have evolved complex responsive and adaptive mechanisms to accommodate Pi deficiency via acquisition, remobilization, and recycling of phosphate so as to maintain Pi homeostasis [6,7,8,9,10]. 

As a result of exposure to Pi deficiency, plants may modify root hair elongation and density [11], and adjust the physiological characters, e.g., the enzymatic activities of peroxidase (POD), superoxide dismutase (SOD), acid phosphatase (ACP), etc. [12,13,14]. Studies have reported that the molecular mechanisms of low−Pi stress may regulate the expression of transcription factors (TFs), consequently, diversifying the expression of functional genes, including those in phosphate transporters (PHT), e.g., PHT1, PHT2, PHT3, and PHT4 [15]. The transcriptomic response to low−Pi stress were carried out to identify the Pi response−related candidate genes in many species, e.g., *OsSPX1* gene from rice [5], *PmWRKY25* gene from Masson pine [16] and *MdPHT1;4* genes from apples [17], etc. 

Apples are one of the most commercially valuable fruit species in the world, and China ranks first both in acreage and yield. However, its growth is severely damaged by the low−Pi stress, especially in South China since the soil is widely short of available Pi due to acidification [18,19,20]. Apple rootstocks play a crucial regulatory role in adaption to environmental adversity, and multifarious rootstocks give various performances in response to nutrient deficiency [21]. Thereby, application of rootstocks with high tolerance to low−Pi status is the efficient strategy to mitigate the available Pi deficiency. Previously, an elite line, MSDZ 109, selected from *Malus mandshurica*, was obtained, which demonstrated a very strong tolerance to low−Pi stress and was used as apple rootstock in the Guizhou Province, Southwest of China, where it was severely affected by low available Pi stress. However, the molecular mechanism underlying the high tolerance to this stress has been unknown so far, which has obviously limited the exploitation of this germplasm.

To better understand the tolerance roles of apples in response to low available Pi stress, the transcriptome sequencing was carried out, subsequently, the morphological and physiological responses were further characterized so as to identify the candidates of functional gene as well as transcription factors from apple rootstock as subjected to low−Pi stress, which will facilitate the illumination for the roles of this line, as well as be helpful for the breeding of low−Pi stress germplasm in the apple rootstock.

## 2. Results

### 2.1. Morphological Traits to Low Phosphorus Treatment

As exposure to low−Pi stress increased, the leaves became deep green in comparison with those of the stress−free (Figure 1). The status of plant morphological traits was quantified as subjected to the low−Pi treatment. The plant height was decreasing by 7.38% for the 30 days after treatment (DAF) in comparison with that of the stress−free, but no significant change was investigated on the 10 DAF. When subjected to low−Pi treatment, the fresh weight and dry weight demonstrated a gradual decrease, e.g., by 18.37% and 25.45%, respectively, after 30−day exposure to low−Pi treatment (Figure 2b,c). Strangely, the root system was improved under low−Pi stress, for example, the total length and surface area showed a gradually increasing trend, with an obvious increase of 41.86% and 41.50% on the 30 DAF in comparison with that of the stress−free (Figure 2d,e). The average diameter and root volume variously increased by 10.64% and 42.86% on the 30 DAF exposed to low−Pi treatment (Figure 2f,g). Compared with the stress−free, the total number of root tips decreased by 16.43% on the 30 DAF, while there was no significant change on the 20 DAF (Figure 2h). These results suggest that the root system of this line was improved when subjected to low−Pi stress, which substantially reflected the high tolerance to this stress.

### 2.2. Physiological Responses to LowPhosphorus Treatment

Due to exposure to the low−Pi stress, the photosynthetic rate increased by 7% on the 10 DAF, however, it obviously decreased by 27% on the 30 DAF in comparison with that of the stress−free, reflecting that the extension of low−Pi treatment duration provided a negative effect on photosynthetic rate (Figure 2i). Interestingly, when subjected to stress, the elevated CAT activity was remarkably investigated in leaves in comparison with those from the stress−free, although a gradually decreased tendency was obtained with the progress of the stress (Figure 3a). SOD activity was diversely increased, e.g., by 61% after 30−day−exposure to low−Pi treatment (Figure 3b). Additionally, an obvious increase (117%) in POD activity was quantified at the 20 DAF in comparison with that of the stress−free (Figure 3c). Proline is beneficial to plants because it protects them from stress damage; during the low−Pi treatment, proline accumulation increased dramatically (Figure 3d). The ACP activity was consistently higher than that of the stress−free and gave an obvious enhancement, e.g., by 166% after 30−day−exposure to low−Pi treatment (Figure 3e). As expected, the total Pi content decreased in comparison with that of the stress−free (Figure 3f). These findings imply that in the previous cases, the enzyme activities of CAT, SOD, and POD increased as a result of exposure to low−Pi stress, which reflected the high resistance to this stress.

### 2.3. Transcriptome Profiling and Validation of RNA Sequencing Data of M. mandshurica to LowPhosphorus Treatment

Transcriptome sequencing was being used to elucidate the molecular mechanism of *M. Mandshurica* underlying the high tolerance to low−Pi stress. The roots and leaves of *M. Mandshurica* were analyzed on the 10 DAF, 20 DAF, and 30 DAF with three biological replicates by Illumina RNA−seq. RNA sequencing of these libraries generated more than 40.37 million raw reads for each sample. Of these reads, GC content was approximately 47.0% for the libraries. After quality control, 33.34 to 44.14 million clean reads were yielded with more than 97.09% of Q20, and 24.28 to 31.99 million clean reads were mapped to the apple genome (Table 1). The average comparison percentages of the genome and the gene were 74.99% and 70.11%.

To further validate the reliability of DEGs analysis in *M. mandshurica* to low−Pi treatment, 16 genes from roots and leaves were randomly selected for qRT−PCR to verify the RNA sequencing data. The qRT−PCR results were largely compatible with the RNA−seq data, with a high correlation coefficient (R^2^ = 0.90) (Appendix A). The relative expression levels were calculated as 2^−ΔΔCt^. Error bars represent the standard deviation of the relative expression level from three biological replicates. The left x−axis represents the results of qRT−PCR, and the right y−axis represents the results of RNA−seq.

### 2.4. Identification of DEGs to Low Phosphorus Treatment

Differential expression analysis was performed using DEGseq2 between leaves and roots on the 10 DAF, 20 DAF, and 30 DAF, with Q value < 0.05 and |Log_2_Ratio| ≥ 1 as the threshold to judge the significance of the expression difference. To find low−Pi responsive genes that were differently expressed between the low−Pi treatment and the stress−free, such comparisons as LCK10 vs. LLP10, LCK20 vs. LLP20, LCK30 vs. LLP30, RCK10 vs. RLP10, RCK20 vs. RLP20, and RCK30 vs. RLP30 (Appendix A) (Figure 4). Compared with that of the control, a total of 347 DEGs and 6015 DEGs were obtained from the leaves of the line as subjected to low−Pi treatment on the 10 DAF and 30 DAF. However, only 35 DEGs were identified on the 20 DAF (Figure 4a). A total of 2299 DEGs and 2754 DEGs were discovered from the roots of the line from exposure to low−Pi treatment on the 20 DAF and 30 DAF. However, only 37 DEGs were investigated on the 10 DAF (Figure 4b). Additionally, according to the Illumina data, pair−wise comparisons (Appendix A) (LLP10 vs. LLP20, LLP10 vs. LLP30, LLP20 vs. LLP30, RLP10 vs. RLP20, RLP10 vs. RLP30, and RLP20 vs. RLP30) of gene expression among the three stages were performed. In leaves, 7444 DEGs of LLP10 vs. LLP20 were discovered to have different expression patterns, with 3935 genes up−regulated and 3509 genes down−regulated. In total, 2749 DEGs of LLP10 vs. LLP30 were discovered, with 1175 genes up−regulated and 1574 genes down−regulated. 4712 DEGs of LLP20 vs. LLP30 were identified, with 2359 genes up−regulated and 2353 genes down−regulated (Figure 4a). In roots, 4359 genes of RLP10 vs. RLP20 were investigated, with 1395 genes up−regulated and 2964 genes down−regulated. A total of 6050 DEGs of RLP10 vs. RLP30 were acquired, with 2481 genes up−regulated and 3569 genes down−regulated. 2009 DEGs of RLP20 vs. RLP30 were discovered, with 1190 genes up−regulated and 819 genes down−regulated (Figure 4b). The Venn diagram analysis revealed that 490 genes and 274 genes were overlapped in leaves and roots (Figure 4c,d).

### 2.5. Classification, Enrichment and Metabolism Overview Analysis of DEGs 

To gain additional insight into the potential roles that distinguish the responses of *M. mandshurica* to low−Pi treatment, the Gene Ontology (GO) database and the Kyoto Encyclopedia of Genes and Genomes (KEGG) database were executed for each group comparison to sequences of unigenes. GO functional analysis showed that DEGs were involved in a variety of molecular functions (14 sub−categories), cellular components (19 sub−categories),and biological processes (28 sub−categories) in leaves and molecular functions (14 sub−categories), cellular components (17 sub−categories) and biological processes (27 sub−categories) in roots. Several stress−related GO terms were commonly found in both leaves and roots, including “cellular process”, “response to stimulus”, “metabolic process”, “biological regulation”, “membrane”, “catalytic activity”, “transcription regulator activity” and “transporter activity” (Figure 5, Appendix A).

To illuminate the function of differentially expressed DEGs. A total of 15,518 unigenes were mapped into 386 and 381 KEGG database pathways in leaves and roots. The 20 pathways with the most DEGs involved in the response to low−Pi treatment (Figure 6) (Appendix A). “Metabolic pathways” (01100), “Biosynthesis of secondary metabolites” (01110), “MAPK signaling pathway” (04010), “Neurotrophin signaling pathway” (04722), “Toll and Imd signaling pathway” (04620), “NOD−like receptor signaling pathway” (04621) and “Carbon metabolism” (1200) were the most represented pathways in leaves and roots.

The *Malus domestica* was used as the reference genome to assess adaptation to low−Pi treatment in leaves and roots, and Madman soft (3.5.1R2) [22,23] was used to display a pathway overview of DEGs in response. The Mercator4 Tool and MapMan4 were used to compare the unigene sequences of 14,971 DEGs, yielding 11,761 MapMan BIN values and 2160 DEGs matched to this path in leaves. The Mercator4 Tool and MapMan4 were used to compare the unigene sequences of 10,880 DEGs, yielding 8670 MapMan BIN values and 1558 DEGs matched to this path in roots. The resulting file was loaded into the MapMan Image Annotator module to generate the metabolism overview map. The MapMan annotation depicts the overall changes in these DEGs metabolic pathways. Photorespiration, lipid metabolism, and the glycolysis process were mainly annotated to three pathways under low−Pi treatment in *M. mandshurica* (Figure 7).

### 2.6. Expression Profiles of Physiologically Relevant DEGs to Low Phosphorus Treatment

By analyzing the expression level of DEGs involving in antioxidant enzymes, SOD was split into three types based on the metal cofactor employed by the enzyme: copper−zinc SOD (SOD [Cu−Zn]), manganese SOD (SOD [Mn]), and iron SOD (SOD [Fe]) by Grene [24]. As subjected to low−Pi stress, currently, two genes of SOD [Fe], one gene of SOD [Mn], three genes of SOD [Cu−Zn], and one gene of CCS were screened from the transcriptome data from *M. mandshurica*. The SOD [Cu−Zn] genes (LOC103434128, LOC103413973, and 103449311) were up−regulated on the 30 DAF, conversely, SOD [Fe] genes (LOC103431645 and LOC103451457) were down−regulated in comparison with those from the stress−free leaves and roots (Figure 8c). Therefore, up−regulated SOD gene expression can play a positive role in this line of response to low−Pi treatment. Transcriptome analysis identified 14 genes as CAT−related, including one gene of CAT and 13 genes of peroxisomal (PEX). Compared with that of the control, the *MmCAT1* (LOC103445262) gene was up-regulated on the 30th DAF in leaves. A total of 56 differentially expressed POD−related genes were acquired from transcriptome data, including 41 genes of POD, 1 gene of 2−Cys peroxiredoxin BAS1 (PER2 BAS1), 4 genes of peroxiredoxin, 4 genes of L−ascorbate peroxidase (APX), 4 genes of glutathione peroxidase (GPX), and 1 gene of overexpressor of cationic peroxidase (OCP3). The *MmPOD6* (LOC103451919) gene was up−regulated on the 30 DAF in leaves and down−regulated in roots. *MmPOD3* (LOC103442590), *MmPOD4* (LOC103437812), *MmPOD11* (LOC103451708), and *MmPOD21* (LOC103433277) were up-regulated on the 30 DAF in leaves (Figure 8b). Therefore, physiology−related significantly differential DEGs revealed that low−Pi treatment affects the activities of enzymes and the expression of related genes.

### 2.7. Analysis of Pi Transporter Genes to Low-Pi Stress

From transcriptome data, 32 differentially expressed Pi transporter genes from 6 families were identified, including 5 genes of PHO1, 6 genes of PHT1, 2 genes of PHT2, 6 genes of PHT3, 10 genes of PHT4, and 3 genes of PHT5, as well as 16 genes of acid phosphatase (AP), respectively (Figure 8a). Exposure to low−Pi treatment, *MmPHT1;3* (LOC10344734) and *MmPHT1;9* (LOC103428150) were strongly induced on the 30 DAF in comparison with that of the stress−free leaves and roots. *MmPHT1;5* (LOC103438596) was strongly up−regulated, a gradually increased tendency was obtained with the progress of stress in roots. *MmPHT1;12* (LOC103404046) was up−regulated on the 10 DAF in leaves and on the 30 DAF in roots compared with that of the stress−free. Additionally, low−affinity Pi transporters of *MmPHT2;1* (LOC103452033) and *MmPHT2;2* (LOC103403064) were weakly up−regulated from exposure to low−Pi treatment on the 30 DAF in both leaves and roots. *MmPHT3;5* (LOC103454899) and *MmPHT4;8* (LOC103422687) were strongly up−regulated in the roots. Additionally, as subjected to stress, the PHO1 family of *MmPHO1;9* (LOC103425833) was strongly down−regulated compared with that of the stress−free, while *MmPHO1;3* (LOC103423874) was strongly up−regulated on the 10 DAF. Moreover, 16 acid phosphatase genes were screened, including *MmPAP3* (LOC103430610), *MmPAP15* (LOC103409175), *MmPAP17* (LOC103438263), *MmPAP20* (LOC103449530), etc. (Figure 8a). *MmPAP15* was strongly up−regulated on the 20 DAF and 30 DAF in roots, while *PAP20* was strongly up−regulated on the 10 DAF in leaves. Therefore, these candidate genes were required to illuminate the molecular mechanism in response to low−Pi treatment of *M. mandshurica*.

### 2.8. Identification of Differentially Expressed TFs to Low Phosphorus Treatment

Results show that a total of 144 TF genes and 130 TF genes were identified in exposure to low−Pi treatment in leaves and roots, respectively. Of these, the 3 most abundant TF families were ethylene response factor (ERF), MYB, and Cys2/His2 (C2H2), while 44 members of the ERF family, 35 members of the MYB family, and 21 members of the C2H2 family exhibited highly expressed patterns of expression after exposure to low−Pi treatment. Venn diagram analysis of the three datasets suggested that 10 genes and 5 genes overlapped in leaves and roots, respectively (Figure 9a,b). Interestingly, among the TFs, according to a heat map analysis, *MmWRKY6* (LOC103452669) and *MmWRKY9* (LOC103433719) were up−regulated as subjected to low−Pi on the 30 DAF in leaves (Figure 9c). *MmMYB6* (LOC103401412) and *MmMYB53* (LOC103433576) were strongly up−regulated on the 30 DAF in roots. In contrast, *MmMYB5* (LOC103441079) and *MmMYBR2* (LOC103424517) were down−regulated in comparison with the stress−free roots. However, the mechanism of its regulation remains to be further researched (Figure 9c). Furthermore, phosphate starvation response (PHR) belongs to the MYB family and contains a highly conserved MYB−DNA binding domain and a CC domain. It is a key regulatory component of the stress−response to Pi deficiency. In the line, *MmPHR1* (LOC103434216) was down−regulated in comparison with that of the stress−free on the 30 DAF in both leaves and roots, whereas *MmPHR2* (LOC103432598) was up−regulated on the 30 DAF in roots. TFs of other families were obtained as they were subjected to low−Pi treatment (Figure 9c). Therefore, it can be speculated that the genes of TFs play an important role in regulating the low−Pi−responsiveness of *M. mandshurica.*

### 2.9. PSI and PSII System Genes in Response to Low Phosphorus Treatment

Pi deficiency inhibits the photosynthetic phosphorylation pathway, which has an indirect effect on photosynthetic activity. In chloroplasts, PSII and PSI−mediated linear electron transport simultaneously produce ATP and NADPH. When treated with low−Pi, the majority of the encoded antenna protein and electron transport were suppressed (Figure 10a,c). In PSI and PSII, the light−responsive genes LHC, PSB, and PSA were almost all down−regulated in comparison with those of the stress-free. According to transcriptome data, 66 differentially expressed photosynthetic response−related genes were screened in leaves. Three metabolic pathways have been linked to these genes: photosynthesis−antenna protein (ko00196), photosynthesis (ko00195), and photosynthetic biological carbon fixation (ko00710). The expression of most genes was down−regulated in LCK10 vs. LLP10 (Figure 10c). The light−harvesting chlorophyll a/b (Chla/b) binding protein is the light−carrying protein of the photosystem II (PS II) light−harvesting complex. In the line, the annotated DEGs of Chl a/b binding protein complex I and II (LHC), namely four genes encoding Chl a/b binding protein complex I (*LHCA1*, *LHCA3*, *LHCA5*, *LHCA6*, *LHCA151*) and five genes encoding Chla/b binding protein complex II (*LHCB*, *LHCB4;1*, *LHCB4;2*, *LHCB4;3*, *LHCB5*) were down−regulated in LLP20 vs. LLP30 (Figure 10b,c). 

## 3. Discussion

### 3.1. Characteristics of Resistance to Low Phosphorus Stress in M. mandshurica

Previous studies have shown that genotypic adaptation to Pi deficiency allows for changes in root architecture [11,25,26]. Low−Pi availability might reprogram root characters, for example, the root length, root branch, root hair, and root density were enhanced, which were documented in wheat [27], *Brassica nigra* [28], soybean [29], and rice [30,31]. In the current cases, the similar root configuration was also obtained in the elite line of *M. mandshurica*. As exposure to low−Pi stress, the increased number of lateral roots contributed to the significant increase in surface area and total root length. On the other hand, low−Pi inhibited the photosynthetic rate, leading to the reduction in the growth and biomass augmentation of shoots, which was also documented in apple [32]. Interestingly, currently, the root system of this line was improved as subjected to low−Pi stress, which substantially reflected the high tolerance to this stress in comparison to apple since the latter is generally regarded as sensitive to low−Pi stress. Additionally, some internal root development−related genes could be regulated during this process in *M. mandshurica.*


Plants can use ROS scavenging enzymes such as SOD, POD, and CAT, etc. to accommodate to adversities including low−Pi stress [33,34,35]. In the present cases, the enzymatic activities of CAT, SOD, and POD obviously increased from exposure to low−Pi stress. However, these physiological responses of this line were not consistent with those of apple under low−Pi stress [32], which may be probably ascribed to the species differences. Furthermore, it was thought to accelerate the utilization of Pi by plants with an increase in ACP activity [13,14]. Simultaneously, although the gene expression patterns in growth status to low−Pi stress have been reported in many species, few studies over how enzyme activity−related genes participate in their regulatory mechanisms have been performed in apple rootstocks. In the present case, the expression levels of those scanvenger genes were similar to the enzyme activity levels. In particular, enzyme activity−related genes of SOD and POD were strongly up−regulated from exposure to low−Pi treatment on the 30 DAF in leaves. Higher levels of SOD, POD, and CAT activities were also great indicators of ROS neutralizing effects, which were effective in scavenging ROS under low−Pi stress. As a result, reduced Pi tolerance might well be connected to efficient ROS scavenging enzymes. This enabled the high tolerance elite germplasm to overcome stress and support growth [35,36,37].

### 3.2. Antioxdative Genes Co−Ordinate the Low Phosphorus Tolerance in M. mandshurica

A tonnage of available evidences has demonstrated that total Pi content was dramatically reduced in plants grown under low−Pi stress [38,39]. Additionally, to alleviate the effects of low−Pi stress, plants might increase Pi absorption and internal phosphorus cycling by modulating the expression of molecular genes such as PHT genes in response to Pi signals [9]. PHT1 is a high−affinity Pi transporter that is essential for maintaining Pi levels in low−Pi conditions [10]. Up to now, some members of the PHT1 family, e.g., *MtPHT1;2* [40], *TaPHT1;3* [41], *GmPHT1;5* [42], *HvPT1;5* [43], *StPHT1;3* [44], *ZmPHT1;7* [45], *PtPHT1;10* [46], *OsPHT1;4* [47] and *MdPHT1;1* [17], were reported to be induction and response by low−Pi treatment. Studies show that *AtPht1;5* is responsible for transporting Pi to the sink organs [48], as well as *AtPht1;8* and *AtPht1;9* are responsible for transporting and transferring Pi from roots in *Arabidopsis* [49]. Interestingly, analysis of transcriptomic data showed that *MmPHT1;3*, *MmPHT1;5*, and *MmPHT1;9* exhibited a greater than 2−fold change in expression to low−Pi treatment in the roots of *M. mandshurica*. Therefore, it is speculated that elite germplasm is linked to a strong tolerance to low−Pi stress. Additionally, to regulate the balance of Pi homeostasis, a set of low−affinity Pi transporters, including *MmPHT2;1*, *MmPHT2;2*, *MmPHT3;5*, and *MmPHT4;8*, has been developed, regulating expression patterns that vary significantly with exposure to the low−Pi treatment in *M. mandshurica*. For example, *PvPht2;1* enhances Pi transport to the chloroplasts in *Arabidopsis* [50]. Whereas, *TaPHT2;1* and *OsPHT2;1* reduced phosphorus accumulation, plant growth, and photosynthetic rates, which were induced by Pi starvation [51,52]. However, few reports of how PHTs participate in low−Pi regulatory mechanisms have been performed in the apple rootstock of *M. mandshurica*. 

### 3.3. MYB TFs Involved in Low Phosphorus Tolerance in M. mandshurica

TFs have been justified to play a vital role in the Pi starvation response via gene expression regulation and signal transmission [53]. The MYB family has been extensively documented among the TFs that respond to low−Pi stress in plants. For example, so far, some members of the MYBs, *AtMYB2* [54], *GmMYB48* [55], *AtPHR1* (*Arabidopsis*) [56], *TaPHR1* (*Triticum aestivum*) [57], *OsPHR2* (*Oryza sativa*) [58], and *FvPHR1* (*Fragaria vesca*) [59], were reported to improve the absorption of Pi. Studies confirmed that overexpression of *NtMYB12* increased total Pi concentration and increased expression of PHT1 family genes (*NtPHT1;1* and *NtPHT1;2*) in tobacco tolerance to low−Pi stress [60]. Previous studies have shown that GmPHR1/4 proteins bind directly to the promoter P1BS elements of their target transporter genes *GmPHT1;1*, *GmPHT1;4* and *GmPHT1;1* [61]. Furthermore, in maize, *ZmPHR1* can bind to the conserved binding site (P1BS) of *ZmGPX*−*PDE1* and *ZmGPX*−*PDE5* promoters and decompose glycerophosphodiester into glycerol−3−phosphate (G3P), indicating that they are involved in the recovery function of Pi from glycerophosphodiester [62]. However, few reports about apple rootstock MYBs exist, except that *MdMYB2* was found to be overexpressed in apples, causing a significant increase in PSI gene expression in response to Pi deficiency [63]. By analyzing the expression of the MYB family related to low−Pi treatment, a total of 35 known MYBs were identified. In terms of expression, *MmMYB5*, *MmMYB6*, *MmMYB53*, and *MmMYBR2* were highly expressed in leaves and roots, indicating that these MYBs have essential roles in various tissues in response to low−Pi treatment of *M. mandshurica*. Interestingly, our research found that two PHRs were stronger expressed in leaves and roots to low−Pi treatment of *M. mandshurica*. However, their regulatory roles are unknown in apple rootstocks. We speculate that *MmPHR1* and *MmPHR2* may be involved in the absorption and transportation of Pi. Additionally, some of available evidences has demonstrated that the *PHL* (PHR1−LIKE) MYB transcription factors are thus likely candidates as targets of *SIZ1*, a SUMO E3 ligase, in the coordination of root developmental responses to low−Pi conditions [64], and the loss of SIZ1 could hampered the responses caused by ROS during low−Pi stress [65,66]. 

## 4. Materials and Methods

### 4.1. Plant Materials and Low Phosphorus Treatments

The apple rootstock of *Malus mandshurica* (Maxim.) Kom (*M. mandshurica*), MSDZ 109, grown in Weining County, Guizhou Province, China (E: 104.12, N: 27.25) was used as the material, and the seeds were collected during maturity. The seeds were generally stratified at 0−4 °C for 45 days until germination, and then cultured with sterile vegetative soil for 3 months. Selected seedlings with a uniform growth trend were hydroponically experimented on according to the modified Hoagland solution [67]. Namely: KNO_3_ 6.0 × 10^−3^; NH_4_NO_3_ 0.5 × 10^−3^; MgSO_4_ 2.0 × 10^−3^; C_10_H_12_N_2_NaFeO.153 × 10^−3^; KI 0.5 × 10^−5^; H_3_BO_3_ 0.1 × 10^−3^; MnSO_4_ 0.148 × 10^−3^; ZnSO_4_ 0.534 × 10^−4^; NaMoO_4_ 0.103 × 10^−5^; CuSO_4_ 0.157 × 10^−6^; COCl_2_ 0.253 × 10^−6^; Ca(NO_3_)_2_ 5.759 × 10^−3^. Before treatment, the MSDZ was pretreated for 3 days with 1/2 of the basic nutrient solution, then treated with low phosphorus (low−Pi, LP). Based on the previous work, 32 µmol/L KH_2_PO_4_ (LP) and 2000 µmol/L KH_2_PO_4_ (CK) were used as the stress and controls, respectively. The seedlings were cultured in a black plastic culture pot with 2000 mL of complete nutrient solution added to each pot and supported by a punched black plastic plate. A total of 50 seedlings were employed to carry out each treatment with three replications. The cultivation conditions were 400−500 µmol·m^−2^·s^−1^ (12 h·d^−1^), 28 °C day and night, and 60–75% relative humidity. The nutrient solution was renewed every 3 days and adjusted to pH 6.5 ± 0.1 with 0.1 mol·L^−1^ NaOH or 0.1 mol·L^−1^ HCl. We rinsed the nutrient pot with deionized water before each change of nutrient solution. The tissues (leaves and roots) of low−Pi treatment (LLP10, LLP20, LLP30, RLP10, RLP20, and RLP30) and control (LCK10, LCK20, LCK30, RCK10, RCK20, and RCK30) were collected on the 10 DAF, 20 DAF, and 30 DAF of *M. mandshurica*, with some of them morphologically examined and physiological indicators measured. Others were frozen in liquid nitrogen right away and held at −80 °C until RNA extraction.

### 4.2. Determination of Morphological and Physiological Parameters 

Following the approach, 15 seedlings were chosen for each treatment, and WinRHIZO root analysis software (Regent Instruments Inc., Quebec, QC, Canada)was used to analyze the root morphological traits after washing with water. The plant height of the above−root sections was measured. The fresh weight of the line was weighed, then dried at 80 °C for 30 min after being heated at 105 °C for 30 min. The dry weight was weighed and statistically assessed. Furthermore, to determine the SOD activity (EC 1.15.1.1) was quantified by monitoring the inhibition of photochemical reduction of nitro−blue tetrazolium (NBT) at 560 nm according to the method described [68]. The CAT activity (EC 1.11.1.6) was determined using a colorimetric assay based on the yellow complex with molybdate and H_2_O_2_, which was measured at 240 nm according to the method described by Goth [69]. The POD activity (EC 1.11.1.7) was measured at 470 nm according to a reported POD assay [70]. The ACP activity was measured at 510 nm as described previously [71]. Proline content was determined at 520 nm as the protocol described by the Bates method [72]. In a nutshell, the fresh leaves of the low−Pi treatment and control were weighed at 0.1 g on the 10 DAF, 20 DAF, and 30 DAF of *M. mandshurica*, and the extract volume was 1:5−10. The leaves were harvested and homogenized in liquid nitrogen, and enzymes were extracted by extraction buffer, Subsequently, placed on ice for testing after the extraction was centrifuged at 8000× *g* for 10 min at 4 °C. The products were determined by an auto−microplate reader (Thermo Fisher Scientific, Waltham, MA, USA). All were determined using kits, which were purchased from Suzhou Comin Biotechnology (Suzhou Comin Biotechnology Co., Ltd., Shuzhou, China). The experimental operation was carried out in strict accordance with the product instructions. The photosynthesis rate (Pn) of new fully expanded leaves of *M. mandshurica*, low−Pi stress on 10 DAF, 20 DAF, and 30 DAF, was quantified using a portable photosynthesis open system (Li−6800, Li−COR Inc., Lincoin, USA) [73]. The measurements were performed under the following conditions: light intensity, 400 µmol m^−2^ s^−1^; leaf temperature 26 ± 2 °C; and relative humidity 60–75%.

### 4.3. Quantification of Plant Total Phosphorus Content

The total phosphorus content was measured at 660 nm according to the described by the molybdenum blue colorimetry method [74]. Briefly, the total phosphorus was digested and converted into inorganic phosphorus. Molybdenum blue and phosphate radicals generate substances with characteristic absorption peaks at 660 nm. The fresh leaves (0.1 g) of *M. mandshurica* seedlings with or without low−Pi treatment on the 10 DAF, 20 DAF, and 30 DAF were harvested and ground in liquid nitrogen. The extract volume was 1:5−10, and enzymes were extracted by extraction buffer. Subsequently, they were placed on ice for testing after the extraction was centrifuged at 10,000× *g* for 10 min. The products were determined by an auto−microplate reader (Thermo Fisher Scientific, Waltham, MA, USA). The content of inorganic phosphorus can be calculated by measuring the light absorption, and then the calculate the total phosphorus content in the leaves. The total phosphorus content of leaves was assayed by a determination kit (Suzhou Comin Biotechnology Co., Ltd., Shuzhou, China). The experimental operation was carried out in strict accordance with the product instructions. 

### 4.4. RNA Extraction and Illumina Sequencing

In total, 36 samples (two conditions (LP and CK) × three stages (10 DAF, 20 DAF and 30 DAF) × two tissues (leaves and roots) × three biological replicates) of *M. Mandshurica* were collected, and the total RNA from leaves and roots at duration treatment was extracted using the TRIZOL kit according to the use method of the manufacturer (Thermo Fisher Scientific, Waltham, MA, USA). The mRNA library was established and sequenced using 36 samples (LLP, RLP, LCK, and RCK) that were mixed from three biological replicates each. A Nano Drop and an Agilent 2100 bioanalyzer were used to qualify and quantify total RNA (Thermo Fisher Scientific, Waltham, MA, USA). The library was validated on the Agilent Technologies 2100 bioanalyzer for quality control. The final library was amplified with phi29 (Thermo Fisher Scientific, Waltham, MA, USA) to make DNA nanoballs (DNBs) that had more than 300 copies of one molecular. DNBs were loaded into the patterned nanoarray and single−end 50−base reads were generated on the BGISEQ500 platform (BGI, Shenzhen, China).

### 4.5. RNA-Seq Data Processing and DEGs Analysis

Filter the sequencing data with SOAPnuke (v1.5.2) [75] and map clean reads to the reference genome with HISAT2 (v2.0.4) [76]. The clean reads were then aligned to the reference coding gene set using Bowtie2 (v2.2.5) [77], and the expression level of each gene was determined using RSEM (v1.2.12) [78]. DESeq2 (v1.4.5) [79] was used to perform differential expression analysis with a Q value of 0.05. Phyper (https://en.wikipedia.org/wiki/Hypergeometric distribution (accessed on 6 March 2020)) based on the Hypergeometric test performed GO (http://www.geneontology.org/ (accessed on 24 May 2020)) and KEGG (https://www.kegg.jp/ (accessed on 10 June 2020)) enrichment analysis of annotated distinct expressed genes to obtain insight into the alteration of phenotype. 

### 4.6. Quantitative Real−Time PCR (qRT−PCR) Analysis

To verify the reliability of RNA−seq results, PowerUp SYBR Green Master Mix (Thermofisher, Chongqing, China) was used for qRT−PCR verification. Sixteen differentially expressed genes induced by low−Pi treatment were randomly selected from the treated roots and leaves for experimental validation (Appendix A). Total RNA was isolated from young leaves and roots using the RNAiso Plus reagent (TaKaRa, Dalian, China) and treated with DNase I (TaKaRa, Dalian, China). The actin genes (β−Actin) of the apple were used as the reference gene. The qRT−PCR amplification was performed as follows: 95 °C for 30 s, followed by 35 cycles of 95 °C for 5 s and 60 °C for 30 s. qRT−PCR specific primers were designed by Primer Premier 5.0 software (Premier Biosof International, Quebec, QC, Canada). The relative expression levels of genes were calculated using the 2^−∆Ct^ method. All validations were performed in three biological and technical replicates.

### 4.7. Statistical Analysis

The parameters were analyzed by variance analysis for statistical testing, and the mean comparison was performed by the Duncan test (*p* < 0.1). The Tukey’s test was performed using the SPSS 21.0 statistical software package. All data are represented by at least three repeated means and a standard deviation (SD). All graphs are used in the drawing software Origin 9.0.

## 5. Conclusions

In conclusion, dramatic morphological, physiological, and transcriptomic changes occurred during exposure to low−Pi stress. The morphological and physiological profiles obviously increased, e.g., total root length, surface area, etc. and the enzymatic activities of CAT, SOD, POD, etc. were in excellent accordance with the enriched pathways of photosynthesis, plant hormone signal transduction and MAPK signaling pathway from the transcriptomic data. Additionally, several enzyme-related candidate genes and low−Pi−responsive genes, e.g., *MmCAT1*, *MmSOD1*, *MmPOD21*, *MmPHT1;5*, *MmPHO1*, *MmPAP1*, etc., were also obtained since their expression status varied among the exposure times, which probably notifies the candidates involved in low−Pi−responsive tolerance in this line. Simultaneously, among the TFs, *MmWRKY6*, *MmWRKY9*, *MmMYB53*, and *MmPHR2* were up−regulated in the line. (Figure 11). The tolerance of *M. mandshurica* to low−Pi stress is inextricably linked to its inherent molecular regulatory system. It shows that the response of *M. mandshurica* to low−Pi stress is a complex procedure and these results provide a broader and better understanding of the Pi response of potential candidate genes. The findings reported herein increase our understanding of the molecular characteristics of *M. mandshurica* response to low−Pi treatment, which contributes to the identification and application of excellent apple rootstock germplasm with low−Pi environmental adaptation.

## Figures and Tables

**Figure 1 ijms-23-04896-f001:**
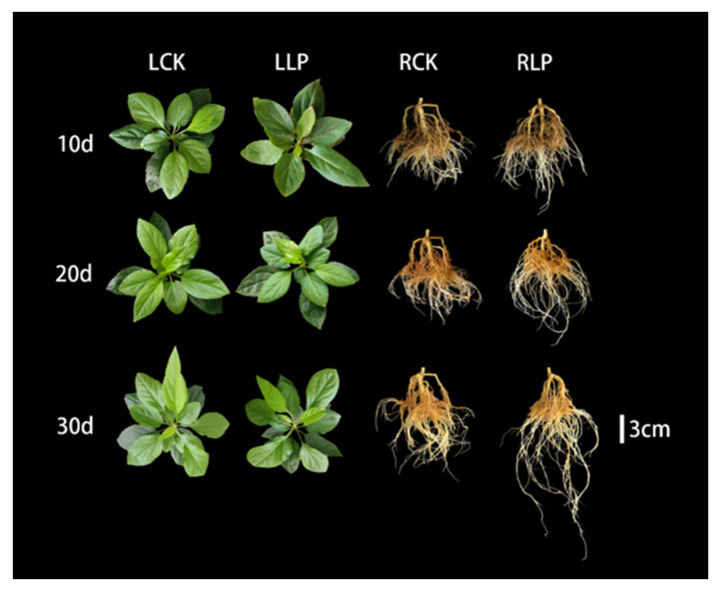
Morphological changes of leaves and roots when exposed to low−Pi treatment for different durations. (LCK: control leaves; RCK: control roots; LLP: Leaves under low−phosphorus treatment; RLP: Roots under low−phosphorus treatment).

**Figure 2 ijms-23-04896-f002:**
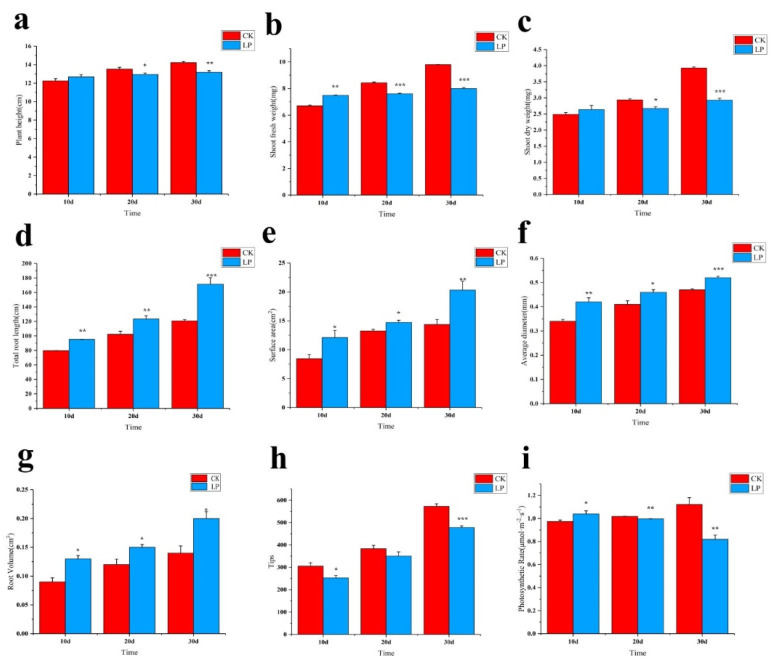
Effects of low−Pi treatment on growth parameters and photosynthetic rate of *M. mandshurica.* CK: control; LP: low−phosphorus treatment. (**a**) Plant height, (**b**) Shoot fresh weight, (**c**) Shoot dry weight, (**d**) Total root length, (**e**) Surface area, (**f**) Average diameter, (**g**) Root volume. (**h**) Tips. (**i**) Photosynthetic rate. The data were averaged from 15 seedlings, and an average (±standard error) was given for each parameter. According to the Tukey’s test, *** *p* < 0.01; ** *p* < 0.01−0.05; * *p* < 0. 1−0.05. The x−axis represents the processing time, and the y−axis represents the content of the measurement index. (All of the following are the same).

**Figure 3 ijms-23-04896-f003:**
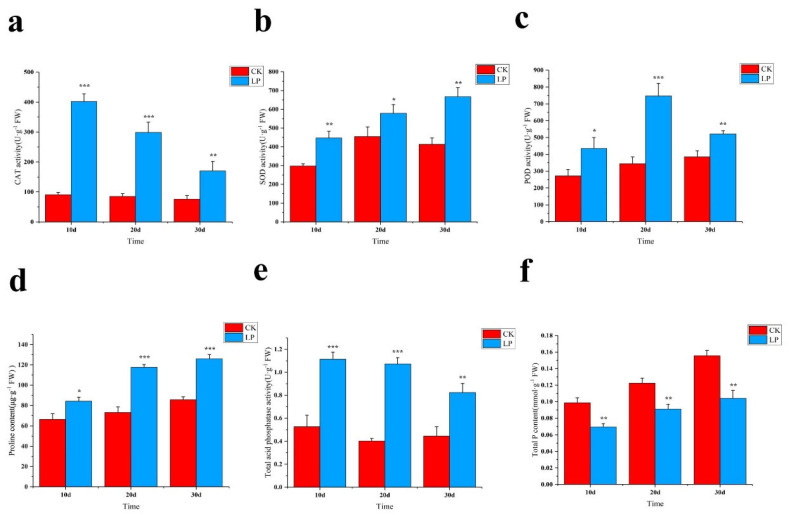
The changes of *M. mandshurica* leaves enzyme activity to low−Pi treatment. (**a**) CAT content, (**b**) SOD content, (**c**) POD content, (**d**) Proline content, (**e**) Acid phosphatase content, (**f**) Total phosphorus content. According to the Tukey’s test, *** *p* < 0.01; ** *p* < 0.01−0.05; * *p* < 0. 1−0.05. The x−axis represents the processing time, and the y−axis represents the content of the measurement index.

**Figure 4 ijms-23-04896-f004:**
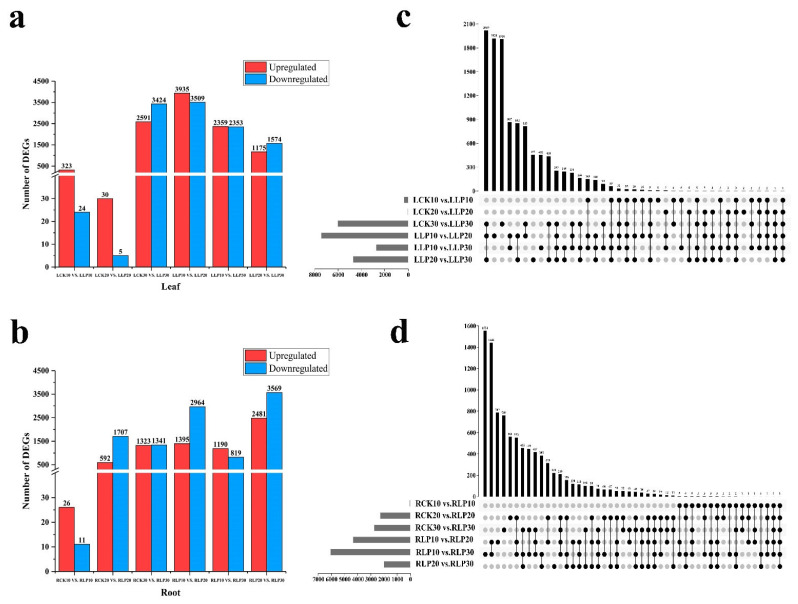
DEGs of under low−Pi stress. Statistics of up− and down−regulated DEGs in leaves (**a**) and roots (**b**) for each pairwise comparison. Venn diagrams of DEGs for control and low−Pi are in leaves (**c**) and roots (**d**). Red represents the DEGs number that is up−regulated, and blue represents the DEGs number that is down−regulated. The x−axis represents the difference comparison scheme for each group, and the y−axis represents the number of corresponding differential genes (DEGs).

**Figure 5 ijms-23-04896-f005:**
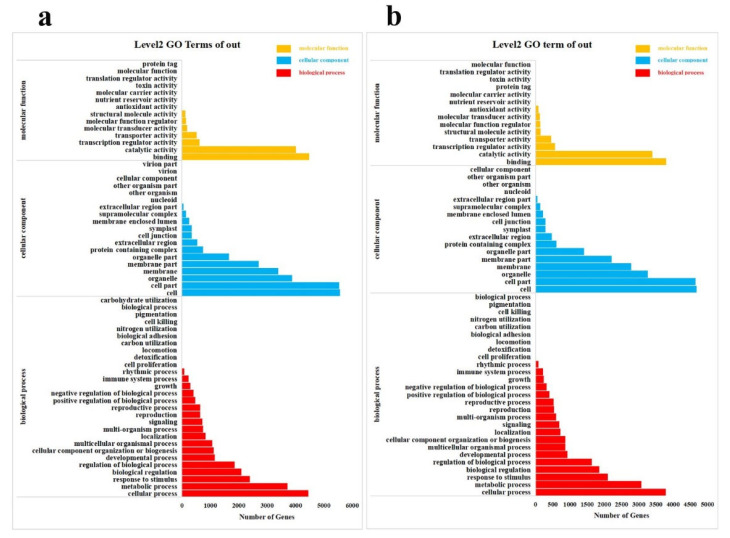
GO classification of DEGs. (**a**) Leaves; (**b**) Roots. The x−axis is the number of genes in the annotation, and the y−axis represents GO categories of gene function.

**Figure 6 ijms-23-04896-f006:**
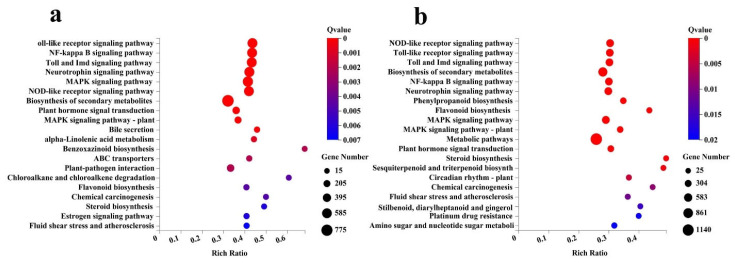
Enrichment of the KEGG annotation of the DEGs. (**a**) Leaves. (**b**) Roots. The x−axis represents the percentage of DEGs belonging to the corresponding pathway, and the y−axis represents the top 20 pathways. The sizes of bubbles represent the number of DEGs in the corresponding pathway, and the colors of the bubbles represent the enrichment Q−value of the corresponding pathway.

**Figure 7 ijms-23-04896-f007:**
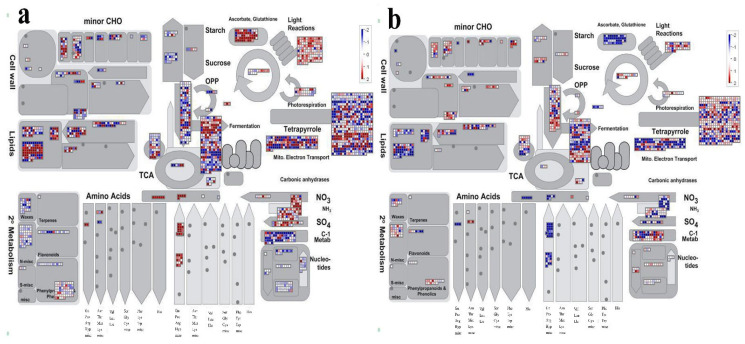
Distribution of up− (in red) and down− (in blue) regulated genes in metabolic pathways in leaves (**a**) and roots (**b**) under low−Pi treatment.

**Figure 8 ijms-23-04896-f008:**
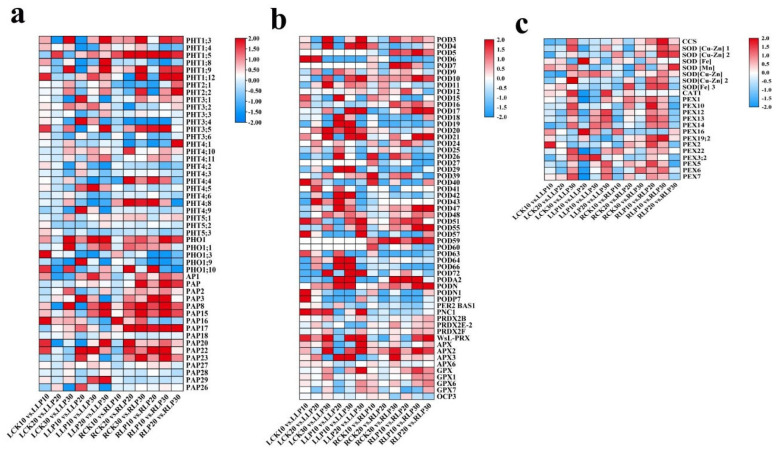
Heat map of DEGs related to phosphorus transport (**a**) and enzymes (**b**,**c**) under low−Pi treatment. The color red represents up−regulated expression, blue represents down−regulated expression, and the white represents that the value of log_2_ (FPKM+1) was zero.

**Figure 9 ijms-23-04896-f009:**
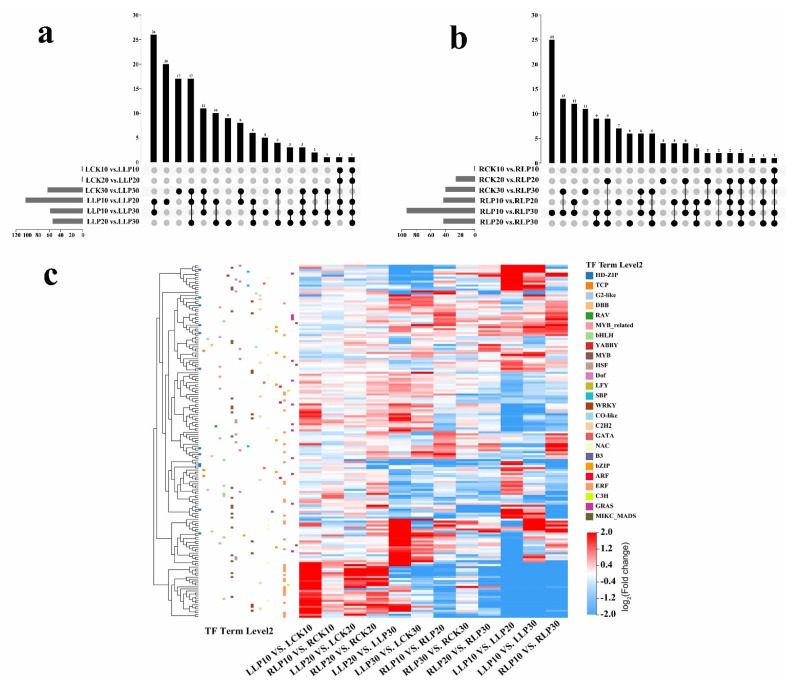
Analysis of differentially expressed TFs in *M. mandshurica* under low−Pi treatment. Venn diagram of (**a**) Leaves and (**b**) Roots. (**c**) Heat map of differential expression of TFs.

**Figure 10 ijms-23-04896-f010:**
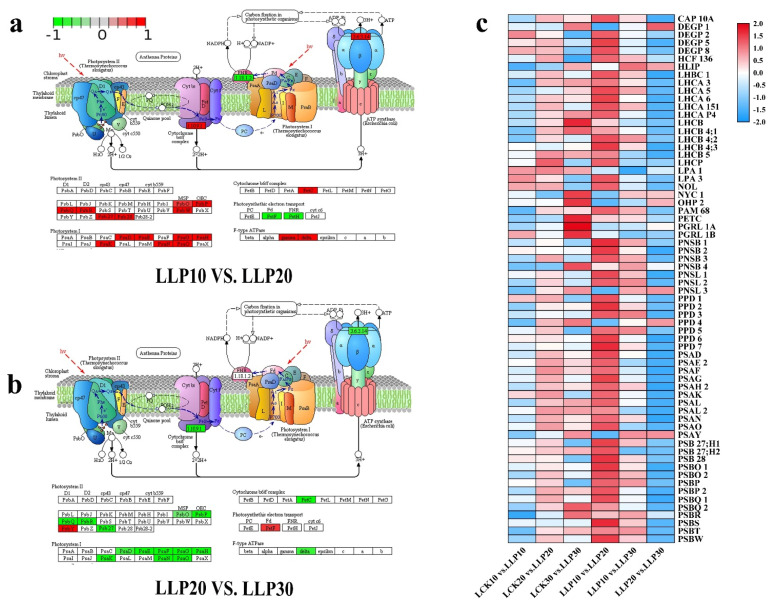
DEGs involved in the photosynthesis pathway of *M. mandshurica* leaves under low−Pi treatment. (**a**) LLP10 vs. LLP20 and (**b**) LLP20 vs. LLP30. (**c**) Heat map of photosynthesis related DEGs. Red represents up−regulated, blue represents down−regulated, and white represents the value of log_2_ (FPKM+1) was zero.

**Figure 11 ijms-23-04896-f011:**
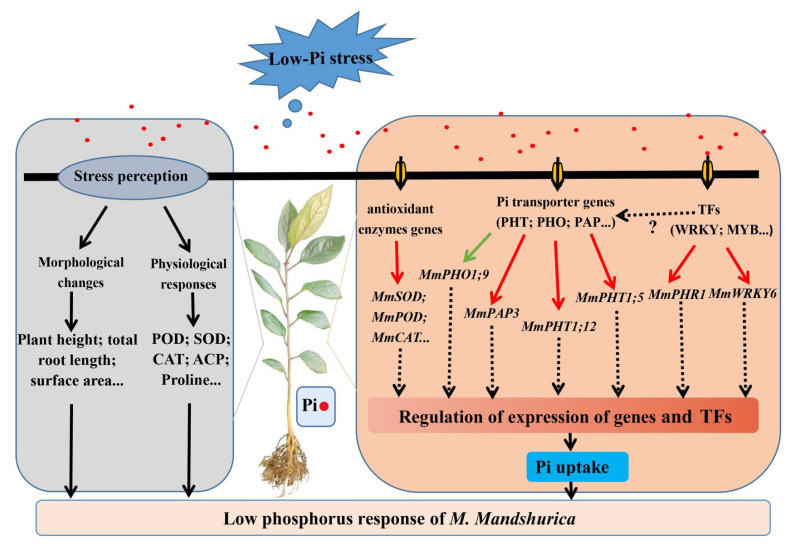
The response model of *M. mandshurica* to low−Pi stress. The red or green lines represent positive or negative effects, respectively; and the black or black dotted lines represent the progressive relationships or the unknowns, respectively.

**Table 1 ijms-23-04896-t001:** Sequencing quality of *M. mandshurica* under low phosphorus treatment.

Sample	Total Raw	Total Clean	Clean Reads	Q20%	Genome Total	Gene Total
Reads (M) Reads (M) (%)				Mapping (%) Mapping (%)		
LCK	46.23	44.15	95.59	98.16	83.67	79.10
LLP	46.41	44.09	95.06	97.99	83.56	79.36
RCK	46.60	44.38	95.28	98.21	71.08	64.78
RLP	45.72	44.50	97.34	98.03	66.47	61.61

## Data Availability

Not applicable.

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
