# Peer review of "Cross−Talk between Transcriptome Analysis and Physiological Characterization Identifies the Genes in Response to the Low Phosphorus Stress in Malus mandshurica"

_ijms, 2022, doi:10.3390/ijms23094896_

Round 1

Reviewer 1 Report

The issue addressed in the paper discusses the identification of genes in response to low phosphorus stress in Malus mandshurica, based on the cross-talk between transcriptome analysis and physiological characteristics. First of all, I find that an important topic, compatible with the journal's scope, was considered.

Such studies are partially analysed in literature. It would be worth presenting the state of the art in a broader way. I suggest a more dilligent, comparative description of other scientific research from the literature (for example, it is possible to add a short state of the art comparative analysis report).

Based on the content of the abstract and then the methodological principles, it is not clear: what is the subject, object, temporal and spatial scope of the study. Furthermore, what is the leading hypothesis of the study?

I also recommend several corrections to improve the quality of this paper:

- to precisely define the research scenario (it is very general); needed to clarify the scope of the study and consequently a clear, step-by-step, simple, synthetic research pattern; yes, the methodology is described, but I recommend more precision, as the reader should know how to repeat a similar analysis on this basis (please consistently correct and complete section 4; I recommend that the methodology section of the study be included in the initial part of the paper; on this basis, the authors should explain step by step their further procedure);

- to improve the readability and description of tables and figures (since they are the basis for analysis verification), supplement the history of their description, a clear and not laconic reference in the paper (I recommend including only readable appendices (figures, tables), unreadable (and therefore not understandable) discard);

that is, supplement the discussion and summary conclusions (please complete section 3 - in connection with the results of section 2) .

Please remember that the formulated objectives - find a clear answer in the conclusion of the study. Is this really the way it works?

Does the conclusion answer all the questions posed at the beginning of the paper (expressed in objectives and hypotheses)? Please complete it and also correct it. The conclusion needs to be supplemented (section 5). I also strongly suggest that recommendations for specific, practical, not only general (and not entirely clear) applications of this research shall be provided.

The language of this paper is relatively correct, however some descriptions would benefit from being more concise. I recommend that the authors cooperate with a native speaker to improve the text (language) of the paper.

Author Response

Dear Reviewers:

Thank you for reviewing our manuscript (ijms-1680141) entitled “Cross-talk between transcriptome analysis and physiological characterization identifies the genes in response to the low phosphorus stress in Malus mandshurica”. Based on the valuable comments and thoughtful suggestions proposed by you, we carefully revised our manuscript. All of the comments and suggestions were taken into account as we modified the manuscript. A detailed list of our responses to the comments and the rephrase sentences were listed below for your reference.

Response reviewer

  1. Such studies are partially analysed in literature. It would be worth presenting the state of the art in a broader way. I suggest a more dilligent, comparative description of other scientific research from the literature (for example, it is possible to add a short state of the art comparative analysis report).

Response:Thank you for your kind suggestion, and we have made the corresponding revisions according to your helpful comment.

(1) We gave a comparative presentation among several scientific researches as “The transcriptomic response to low-Pi stress were carried out to identify the Pi response-related candidate genes in many species, e.g. OsSPX1 from rice [5], PmWRKY25 from masson pine [17] and MdPHT1;4 from apple [18] etc.” in lines 55-61 of the revised version.

2) We added a comparison of the obtained results with similar studies on apple in lines 321-326 of the revised version. “On the other hand, low-Pi inhibited the photosynthetic rate, leading to the reduction in the growth and biomass augmentation of shoots, which was also documented in apple [32]. Interestingly, currently, the root system of this line was improved as subjected to low-Pi stress, which substantially reflected the high tolerance to this stress in comparison to apple since the latter is generally regarded as sensitive to low-Pi stress. ”

3) We added a comparison of the obtained results with similar studies on apple at lines 329-333 of the manuscript. “In the presnt cases, the enzymatic activities of CAT, SOD and POD obviously increased as exposure to low-Pi stress. However, these physiological responses of this line were not consistent with those of apple under low-Pi stress [32], which may be probably ascribed to the species differences. ”

  1. Based on the content of the abstract and then the methodological principles, it is not clear: what is the subject, object, temporal and spatial scope of the study. Furthermore, what is the leading hypothesis of the study?

Response:Thank you for your powerful suggestion.

The clue of the present investigation is as following: Previously, an elite line, MSDZ 109, was justified for its excellent tolerance to low-Pi stress. Currently, the physiological responses of this line to low-Pi stress were characterized, and their roots as well as leaves were used to carry out transcriptome analysis, so as to illuminate the potential molecular pathways and identify the genes involved in low-Pi stress response. In combination with the physiological characterization, several low-Pi-responsive genes were identified, and the genes implicated in Pi uptake and transportation as well as transcription factors were also obtained.

The leading hypothesis of the study: as exposure to low-Pi stress, the expressions of many genes were regulated, leading to the alteration of molecular pathways, subsequently enhance low-Pi tolerance. To address this hypothesis, we used the roots and leaves (spatial scope, line 21) of line MSDZ 109 (subject) as exposure to Pi deficiency for different days (temporal scope, line 20) to carry out transcriptome analysis, so as to illuminate the potential molecular pathways and identify the genes involved in low-Pi stress response in this line (objective), which was presented in lines 22-24 of the new version. Results showed many candidate genes e.g. MmCAT1, MmSOD1, MmPOD21 (added in lines 30 of new version), MmPHT1;5, MmPHO1, MmPAP1 (line 34) were up-regulated, leading to the elevated activities of antioxidant enzymes (lines108-120) and attenuated Pi shortage (lines 120-121) etc., consequently give rise to the high tolerance to Pi deficiency.

  1. - to precisely define the research scenario (it is very general); needed to clarify the scope of the study and consequently a clear, step-by-step, simple, synthetic research pattern; yes, the methodology is described, but I recommend more precision, as the reader should know how to repeat a similar analysis on this basis (please consistently correct and complete section 4; I recommend that the methodology section of the study be included in the initial part of the paper; on this basis, the authors should explain step by step their further procedure);

Response:Thank you for your suggestion. We added descriptions of these sections of the new version.

1) We added descriptions of “leaves and roots” in line 416 of the new version.

2) We added descriptions of “at 560 nm according to the method described” in lines 428-429 of the new version.

3) We added descriptions of “CAT activity was determined using the colorimetric assay based on the yellow complex with molybdate and H2O2, which was measured at 240 nm according to the method described by Goth.” in lines 429-431 of the new version.

4) We added descriptions of “at 470 nm” in line 431 of the new version.

5) We added descriptions of “at 510 nm” at lines 432-433 of the new version.

6) We added descriptions of “at 520 nm” in line 433 of the new version.

7) The description of the photosynthetic rate measurement was presented in section Materials and Methods in lines 436-439 of the new version. “Net photosynthesis rate (Pn) was quantified using a portable photosynthesis system (Li-6800, Li-COR Inc., USA) [73]. The measurements were performed under the fol-lowing conditions: light intensity, 400 µmol m−2 s−1; leaf temperature 26±2℃; and relative humidity 60%-75%.”

8) The descriptions of the total phosphorus content in lines 441-444 of the new version. “The total phosphorus content was measured at 660 nm according to the described by the molybdenum blue colorimetry method [74]. The total phosphorus content of tissues was assayed by a determination kit (UV-Vis Spectrophotometry) (Suzhou Comin Biotechnology Co., Ltd.,).”

  1. - to improve the readability and description of tables and figures (since they are the basis for analysis verification), supplement the history of their description, a clear and not laconic reference in the paper (I recommend including only readable appendices (figures, tables), unreadable (and therefore not understandable) discard); that is, supplement the discussion and summary conclusions (please complete section 3 - in connection with the results of section 2) .

Response:We appreciate it very much for this good suggestion, and we have made corresponding changes and checked the full text.

1) We added descriptions of “leaves and roots” to the Figure 1 in line 97 of page 3 of the new version.

2) We added summary conclusions of “These results suggest that the root system of this line was improved as subjected to low-Pi stress, which substantially reflected the high tolerance to this stress.” in lines 95-96 of the new version.

3) We added summary conclusions of “These findings imply that in the previous cases, the enzyme activities of CAT, SOD, and POD increased as a result of exposure to low-Pi stress, which reflected the high re-sistance to this stress.” in lines 121-123 of the new version.

4) We added descriptions of “The left x-axis represents the results of qRT-PCR, and the right y-axis represents the results of RNA-seq.” to Figure S1 in lines 146-147of the new version.

5) We added descriptions and replaced pictures with higher definition of “The x-axis is the number of genes in the annotation, and the y-axis represents GO categories of gene function.” to Figure 5 in lines 200 -201 of page 7 of the new version.

6) As you suggested, we decided to discard the table of Additional file 8: Table S8.

7) We replaced the picture with higher definition and added a description of “Heat map of TFs differential expression in leaves and roots for each pairwise comparison.” to Figure 9 of page 10 of the new version. 

  1. Please remember that the formulated objectives - find a clear answer in the conclusion of the study. Is this really the way it works? Does the conclusion answer all the questions posed at the beginning of the paper (expressed in objectives and hypotheses)? Please complete it and also correct it. The conclusion needs to be supplemented (section 5). I also strongly suggest that recommendations for specific, practical, not only general (and not entirely clear) applications of this research shall be provided.

Response:Thank you for your kind suggestion, and we have made the corresponding revisions according to your helpful comment.

1) We have made the corresponding modification in lines 487-490 of the revised version. “In conclusion, morphological indications and physiological responses such as changes in root system and enzyme activity of M. mandshurica under low-Pi stress. The tolerance of M. mandshurica to low-Pi stress is inextricably linked to its inherent molecular regulatory system.”

2) We added comments and descriptions of these situations at lines 497-499 of the manuscript. “It shows that the response of M. mandshurica to low-Pi stress is a complex procedure and these results provide a broader and better understanding of the Pi response of potential candidate genes. ’’

  1. The language of this paper is relatively correct, however some descriptions would benefit from being more concise. I recommend that the authors cooperate with a native speaker to improve the text (language) of the paper.

Response:Thank you for your valuable and thoughtful comments. We have carefully checked and improved the English writing in the revised manuscript.

We sincerely hope that this revised manuscript has addressed all your comments and suggestions. We appreciated for reviewers’ warm work earnestly, and hope that the correction will meet with approval. Once again, thank you very much for your comments and suggestions.

Reviewer 2 Report

This is very interesting story and enough to be accepted in this journal based on bunch of the data. I just got few things to be addressed by authors.

  1. Summary figure in Conclusion section should be added.
  2. It has been known that phosphorus (Pi) is a macronutrient essential for plant growth, development, and reproduction as general physiologic responses. Therefore, authors should also show effects on these events. 

Author Response

Dear Reviewers:

Thank you for reviewing our manuscript (ijms-1680141) entitled “Cross-talk between transcriptome analysis and physiological characterization identifies the genes in response to the low phosphorus stress in Malus mandshurica”. Based on the valuable comments and thoughtful suggestions proposed by you, we carefully revised our manuscript. All of the comments and suggestions were taken into account as we modified the manuscript. A detailed list of our responses to the comments and the rephrase sentences were listed below for your reference.

Response reviewer

  1. Summary figure in Conclusion section should be added.

Response:We appreciate it very much for this good suggestion, and we have added it, which was shown as Figure 11 in Page 16.

Figure 11. The response model of M. mandshurica to low-Pi stress. The red or green lines represent positive or negative effects, respectively; and the black or black dotted lines represent the progressive relationships or the unknowns, respectively. 

  1. It has been known that phosphorus (Pi) is a macronutrient essential for plant growth, development, and reproduction as general physiologic responses. Therefore, authors should also show effects on these events.

Response:We are very grateful for your suggestions. In the present work, Pi deficiency was transcriptionally proven to highly affects the process of photosynthesis, plant hormone signal transduction and biosynthesis of secondary metabolites. Low-Pi stress gave diverse effect on the expression of genes involved in carbon metabolism in M. mandshurica seedlings, particularly genes related to glycolysis, the tricarboxylic acid (TCA) cycle, and the phosphorylation pathway, which was presented in lines 193 of page 6. Based on the suggestion, we are further investigating the roles of these genes in low-Pi tolerances of this elite line, which will be showed in the consecutive presentation.

We sincerely hope that this revised manuscript has addressed all your comments and suggestions. We appreciated for reviewers’ warm work earnestly, and hope that the correction will meet with approval. Once again, thank you very much for your comments and suggestions.

Reviewer 3 Report

This is an interesting article on the identification of genes in apple rootstock in response to phosphorus starvation, but there are some comments.

L.23. What is DAF? Please provide the meaning of the abbreviation.

L.63-66. … “demonstrated a very strong tolerance to low-Pi stress” …. Was this data published or not?

L.81. "sebjected" should be corrected to "subjected".

L.117. It would be more correct to write leaves, not seedlings.

L.132, 309, 474. Names of plant species should be given in italics.

A comparison of the obtained results with similar studies on apple should be added to the Discussion section (f.e., Sun et al. Transcriptome and metabolome analyzes revealed the response mechanism of apple to different phosphorus stresses. Plant Physiol. Biochem. 2021, 167:639-650).

A description of the photosynthetic rate measurement (L.102-105) should be added to the Materials and Methods section.

The style of references does not meet the requirements of the journal.

Author Response

Dear Reviewers:

Thank you for reviewing our manuscript (ijms-1680141) entitled “Cross-talk between transcriptome analysis and physiological characterization identifies the genes in response to the low phosphorus stress in Malus mandshurica”. Based on the valuable comments and thoughtful suggestions proposed by you, we carefully revised our manuscript. All of the comments and suggestions were taken into account as we modified the manuscript. A detailed list of our responses to the comments and the rephrase sentences were listed below for your reference.

Response reviewer

  1. 23. What is DAF? Please provide the meaning of the abbreviation.

Response:Thank you very much for your suggestions, and sorry for our imprecise presentation. We have added the meaning of the abbreviation as “day after treatment (DAF).” in lines 23-24 of the new version.

  1. 63-66. … “demonstrated a very strong tolerance to low-Pi stress” …. Was this data published or not?

Response:This excellent germplasm with low-Pi tolerance was selected from our previous work, and the morphological and physiological response data has not yet be published so far.

  1. 81. "sebjected" should be corrected to "subjected".

Response:I’m so sorry for our carelessness. We have made the corresponding modification in line 86 of the new version.

  1. 117. It would be more correct to write leaves, not seedlings.

Response:Thank you very much for your suggestions. We have made the corresponding modification in line 124 of the revised version.

  1. 132, 309, 474. Names of plant species should be given in italics.

Response:Thank you very much for your suggestions, and sorry for our imprecise writing. We have corrected in lines 139, 318, 497 of the revised version.

  1. A comparison of the obtained results with similar studies on apple should be added to the Discussion section (e., Sun et al. Transcriptome and metabolome analyzes revealed the response mechanism of apple to different phosphorus stresses. Plant Physiol. Biochem. 2021, 167:639-650).

Response:Thank you for your kind suggestion, and we have made the intensive revision according to your helpful comment.

1) We added a comparison of the obtained results with similar studies on apple in lines 321-326 of the revised version. “On the other hand, low-Pi inhibited the photosynthetic rate, leading to the reduction in the growth and biomass augmentation of shoots, which was also documented in apple [32]. Interestingly, currently, the root system of this line was improved as subjected to low-Pi stress, which substantially reflected the high tolerance to this stress in comparison to apple since the latter is generally regarded as sensitive to low-Pi stress. ”

2) We added a comparison of the obtained results with similar studies on apple at lines 329-333 of the manuscript. “In the presnt cases, the enzymatic activities of CAT, SOD and POD obviously increased as exposure to low-Pi stress. However, these physiological responses of this line were not consistent with those of apple under low-Pi stress [32], which may be probably ascribed to the species differences. ”

In addition, we believe that comparing with apple can better understand the ability to respond to low-Pi stress of M. mandshurica. Simultaneously, it is also beneficial to our further research on apple rootstocks. Thank you again for your suggestions, your suggestions are of great help to us.

  1. A description of the photosynthetic rate measurement (L.102-105) should be added to the Materials and Methods section.

Response:I’m so sorry for our carelessness. The description of the photosynthetic rate measurement was presented in section Materials and Methods in lines 436-439 of the new version. “Net photosynthesis rate (Pn) was quantified using a portable photosynthesis system (Li-6800, Li-COR Inc., USA) [73]. The measurements were performed under the fol-lowing conditions: light intensity, 400 µmol m−2 s−1; leaf temperature 26±2℃; and relative humidity 60%-75%.”

  1. The style of references does not meet the requirements of the journal.

Response:Thank you very much for your suggestions. We have made corresponding changes and checked the full text.

We sincerely hope that this revised manuscript has addressed all your comments and suggestions. We appreciated for reviewers’ warm work earnestly, and hope that the correction will meet with approval. Once again, thank you very much for your comments and suggestions.

Round 2

Reviewer 1 Report

Dear Authors!

Thank you for improving the text. Your corrections are partially acceptable. I believe that you have taken up an important topic, in line with the profile of the journal.
However, please consider the methodological additions. Does the conclusion answer all the questions posed at the beginning of the study (expressed in the objectives and hypotheses)? Please complete this and also correct it. Conclusions need to be supplemented.
Careful editing of your text is also necessary.

Author Response

Dear Reviewers:

Thank you for reviewing our manuscript (ijms-1680141) entitled “Cross-talk between transcriptome analysis and physiological characterization identifies the genes in response to the low phosphorus stress in Malus mandshurica”. Based on the valuable comments and thoughtful suggestions proposed by you, we again carefully revised the manuscript. All of the comments and suggestions were taken into account as we modified the manuscript. A detailed list of our responses to the comments and the rephrase sentences were listed below for your reference.

Response reviewer:

  1. The methodological additions.

Response:We are very grateful for your suggestions. The description of the methodological was presented in lines 453-466, 473-478 and 480-492 of the new version.

Lines 453-466: Furthermore, to determine the SOD activity (EC 1.15.1.1) was quantified by monitoring the inhibition of photochemical reduction of nitro-blue tetrazolium (NBT) at 560 nm according to the method described [68]. The CAT activity (EC 1.11.1.6) was determined using a colorimetric assay based on the yellow complex with molybdate and H2O2, which was measured at 240 nm according to the method described by Goth [69]. The POD activity (EC 1.11.1.7) was quantified at 470 nm according to a reported POD assay [70]. The ACP activity was measured at 510 nm as described previously [71]. Proline content was determined at 520 nm as the protocol described by the Bates method [72]. In a nutshell, the fresh leaves of the low-Pi treatment and control were weighed at 0.1g on the 10 DAF, 20 DAF, and 30 DAF of M. mandshurica, and the extract volume was 1:5-10. The leaves were harvested and homogenized in liquid nitrogen, and enzymes were extracted by extraction buffer, Subsequently, placed on ice for testing after the extraction was centrifuged at 8000 × g for 10 minutes at 4 °C. The products were determined by an auto-microplate reader (Thermo Fisher Scientific, Inc.).

Lines 473-478: The photosynthesis rate (Pn) of new fully expanded leaves of M. mandshurica, low-Pi stress on 10 DAF, 20 DAF, and 30 DAF, was quantified using a portable photosynthesis open system (Li-6800, Li-COR Inc., USA) [73]. The measurements were performed under the following conditions: light intensity, 400µmol m−2 s−1; leaf temperature 26±2℃; and relative humidity 60%-75%.

Lines 480-492: The total phosphorus content was measured at 660 nm according to the described by the molybdenum blue colorimetry method [74]. Briefly, the total phosphorus was digested and converted into inorganic phosphorus. Molybdenum blue and phosphate radicals generate substances with characteristic absorption peaks at 660 nm. The fresh leaves (0.1g) of M. mandshurica seedlings with or without low-Pi treatment on the 10 DAF, 20 DAF, and 30 DAF were harvested and ground in liquid nitrogen. The extract volume was 1:5-10, and enzymes were extracted by extraction buffer, Subsequently, placed on ice for testing after the extraction was centrifuged at 10000 × g for 10 minutes. The products were determined by an auto-microplate reader (Thermo Fisher Scientific, Inc.). The content of inorganic phosphorus can be calculated by measuring the light absorption, and then the calculate the total phosphorus content in the leaves. The total phosphorus content of leaves was assayed by a determination kit (Suzhou Comin Biotechnology Co., Ltd.,). The experimental operation was carried out in strict accordance with the product instructions.

  1. Does the conclusion answer all the questions posed at the beginning of the study (expressed in the objectives and hypotheses)? Please complete this and also correct it. Conclusions need to be supplemented.

Response:Thank you for your powerful suggestion.

In conclusion, dramatic morphological, physiological and transcriptomic changes occurred as exposure to low-Pi stress. The morphological and physiological profiles of obviously increased, e.g. total root length, surface area, etc. and the enzymatic activities of CAT, SOD, POD, etc. were in excellent accordance with the enriched pathways of photosynthesis, plant hormone signal transduction and MAPK signaling pathway from the transcriptomic data. Additionally, several enzyme-related candidate genes and low-Pi-responsive genes, e.g. MmCAT1, MmSOD1, MmPOD21, MmPHT1;5, MmPHO1, MmPAP1, etc., were also obtained since their expression status varied among the exposure times, which probably notifies the candidates involved in low-Pi-responsive tolerance in this line. Simultaneously, among the TFs, MmWRKY6, MmWRKY9, MmMYB53 and MmPHR2 were up-regulated in the line. (Figure 11) (objectives and hypotheses, lines 538-551). The tolerance of M. mandshurica to low-Pi stress is inextricably linked to its inherent molecular regulatory system. It shows that the response of M. mandshurica to low-Pi stress is a complex procedure and these results provide a broader and better understanding of the Pi response of potential candidate genes. The findings reported herein increase our understanding of the molecular characteristics of M. mandshurica response to low-Pi treatment, which contributes to the identification and application of excellent apple rootstock germplasm with low-Pi environmental adaptation.

  1. Careful editing of your text is also necessary.
    Response:Thank you for your kind suggestion, and we will again be careful in our editing of our text according to your helpful comment.

The red highlighted part is the second revision of the manuscript.

We sincerely hope that this revised manuscript has addressed all your comments and suggestions. We appreciated for reviewers’ warm work earnestly, and hope that the correction will meet with approval. Once again, thank you very much for your comments and suggestions.

Reviewer 2 Report

This paper is now acceptable.

Author Response

Dear Reviewer:

Thanks very much for your kind work with regard to our manuscript. On behalf of my co-authors, we would like to express our great appreciation to you.

Thank you and best regards,

Yours sincerely,

Dr. Hong Zhao

Email: zhaohonggzu@163.com
